# Task Scheduling Model of Double-Deep Multi-Tier Shuttle System

**Yanyan Wang [1,\*], Rongxu Zhang [1], Hui Liu [2], Xiaoqing Zhang [1] and Ziwei Liu [1]**

1    Logistics Engineering in School of Control Science and Engineering, Shandong University,
     Jinan 250061, China
2    School of Software, Shandong University, Jinan 250061, China
\*    Correspondence: yanyan.wang@sdu.edu.cn; Tel.: +86-531-88392269

**Abstract:** As a new type of part-to-picker storage system, the double-deep multi-tier shuttle system has been developed rapidly in the e-commerce industry because of its high flexibility, large storage capacity, and robustness. The system consists of a multi-tier shuttle sub-system that controls horizontal movement and a lift sub-system that manages vertical movement. The combination of shuttles and lifts, instead of a stacker crane in conventional automated storage and retrieval system, undertakes inbound/outbound tasks. Because of the complex structure and numerous equipment of the system, task scheduling has become a major difficulty in the outbound operation of the double-deep multi-tier shuttle system. Figuring out methods to improve the overall efficiency of task scheduling operations is the focus of current system application enterprises. This paper introduces the task scheduling problem for the shuttle system. Inspired from workshop production scheduling problems, we minimize the total time of a batch of retrieval tasks as the objective function, applying the modified Simulated Annealing Algorithms (SAAs) to solve the task scheduling problem. In conclusion, we verified the proposed model and the algorithm efficiency, using case studies.

**Keywords:** double-deep multi-tier shuttle system; modified Simulated Annealing Algorithms; task scheduling problem; part-to-picker storage system

---

## 1. Introduction

Double-deep multi-tier shuttle system is a kind of automated storage and retrieval system (AS/RS) based on common double-depth shelf system, which is composed of shuttles and lifts. The aisle of this system contained a lift and multiple shuttles. The shuttle is responsible for the horizontal movement and carrying Stock Keeping Units (SKUs) to the corresponding tier's I/O point of the aisle. The lift is responsible for the movement of SKUs in vertical direction, transporting goods from I/O point to the storage platform. Double-deep multi-tier shuttle system has advantages in space utilization and storage capacity from AS/RS, but also in flexibility. It can meet the demand of multi-variety, high flexibility, and high-density dynamic storage and retrieval in the future manufacturing and distribution environment. Outbound operation is the most important function of multi-tier shuttle system. At present, one of the difficulties faced by multi-tier shuttle system in sorting out operation is how to schedule tasks. The shuttles and lifts in the multi-tier shuttle system is different from the stacker in the traditional AS/RS. The shuttle is responsible for the horizontal movement and the lift is responsible for the vertical movement. It is the key and difficult point to schedule equipment to outbound operation of the multi-tier shuttle system.

Compared with the single-deep multi-tier shuttle system, the double-deep multi-tier shuttle system can improve the storage capacity of logistics enterprises without hindering outbound picking process and the utilization rate of equipment [1]. The system pays equal attention on sorting and

storage, so how to ensure a flexible and efficient outbound picking process under high-density storage state is the key to the system's optimization. The double-deep outbound picking process is shown in Figure 1. Because the multi-tier shuttle system studied in this paper uses double-depth shelves, the dense storage capacity is higher, but the outbound operation will occur rearrangement with a certain probability, which increases the complexity of the system task scheduling. Due to the special layout structure of double-deep position, the problem of task schedule in outbound picking process is the difficulty that restricts the wide range use of double-deep multi-tier shuttle system.

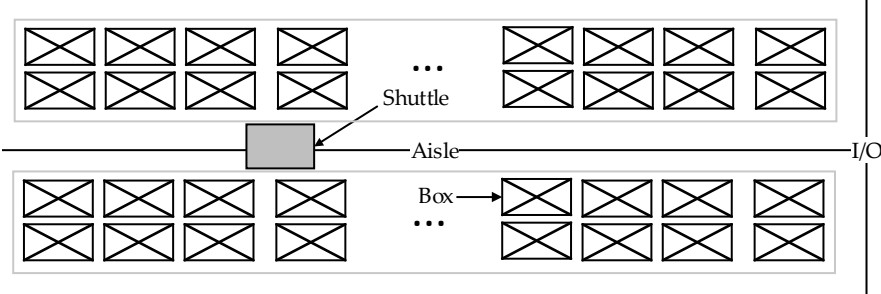

**Figure 1.** Double-deep outbound picking process.

The outbound operation of the double-deep multi-tier shuttle system is shown in Figure 2. There are multiple tiers in an aisle, and there is a corresponding I/O point on each floor. Shuttle is responsible for the horizontal movement of SKUs, taking SKUs to the corresponding I/O point of the tier, and each tier has a shuttle that is responsible for the storage and retrieval. The I/O point of each tier locates at the head end of the shelf, linking the transport operation of shuttle and lift. In addition, the lift corresponds to another I/O. One or two pillars support the lift and the SKU is loaded by the lifting platform which moves vertically.

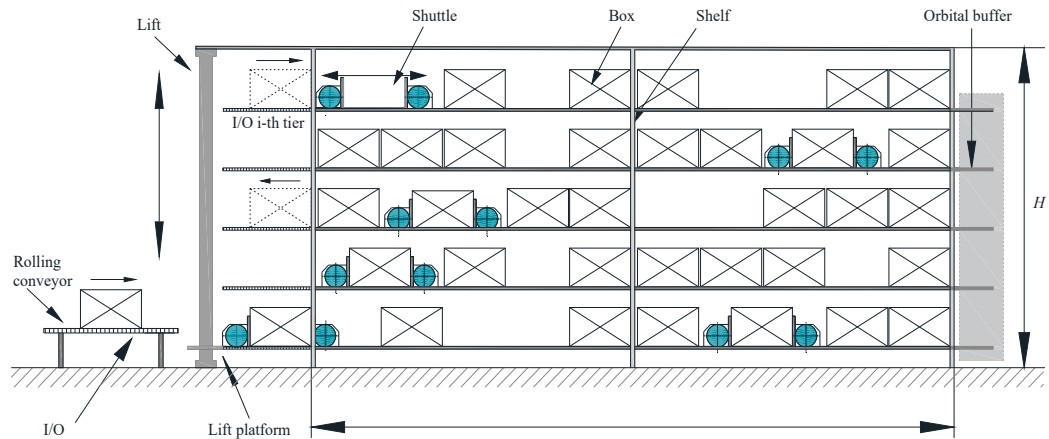

**Figure 2.** Double-deep multi-tier shuttle system.

The double-deep multi-tier shuttle system gives priority to the picking process, and takes storage as the auxiliary part. Therefore, it is significant to research the outbound operation of the system. In the double-deep multi-tier shuttle system, the outbound operation can be divided into two types: Retrieval based on sequence order, and order batching, respectively. Retrieval based on sequence order means to pick needed SKU in sequence according to the arrival time of orders; retrieval based on order batching refers to selecting orders within a certain time window, batching the orders according to certain optimization principles, then combining required SKUs of each batch, finally transferring SKUs out of warehouse and picking them to order. In this way, SKUs in the same batch and different orders can be transferred at one time, which reduces the outbound times of turnover boxes and improves the efficiency of outbound picking. This paper used the second type: Retrieval based on order batching.

The retrieval process that involves rearrangement is shown in Figure 3. Firstly, orders are batched according to certain optimization principles, and then the sequence of outbound tasks of each batch is determined. Next, since each SKU can be stored in multiple locations of the system, it is necessary to determine the location of outbound SKU according to certain principles. Finally, the outbound operation is completed by shuttles and lifts first, and then the conveying equipment transports the turnover box to the workstation for picking. In the retrieval process, the shuttle may have a rearrangement operation with the status of the turnover box's position.

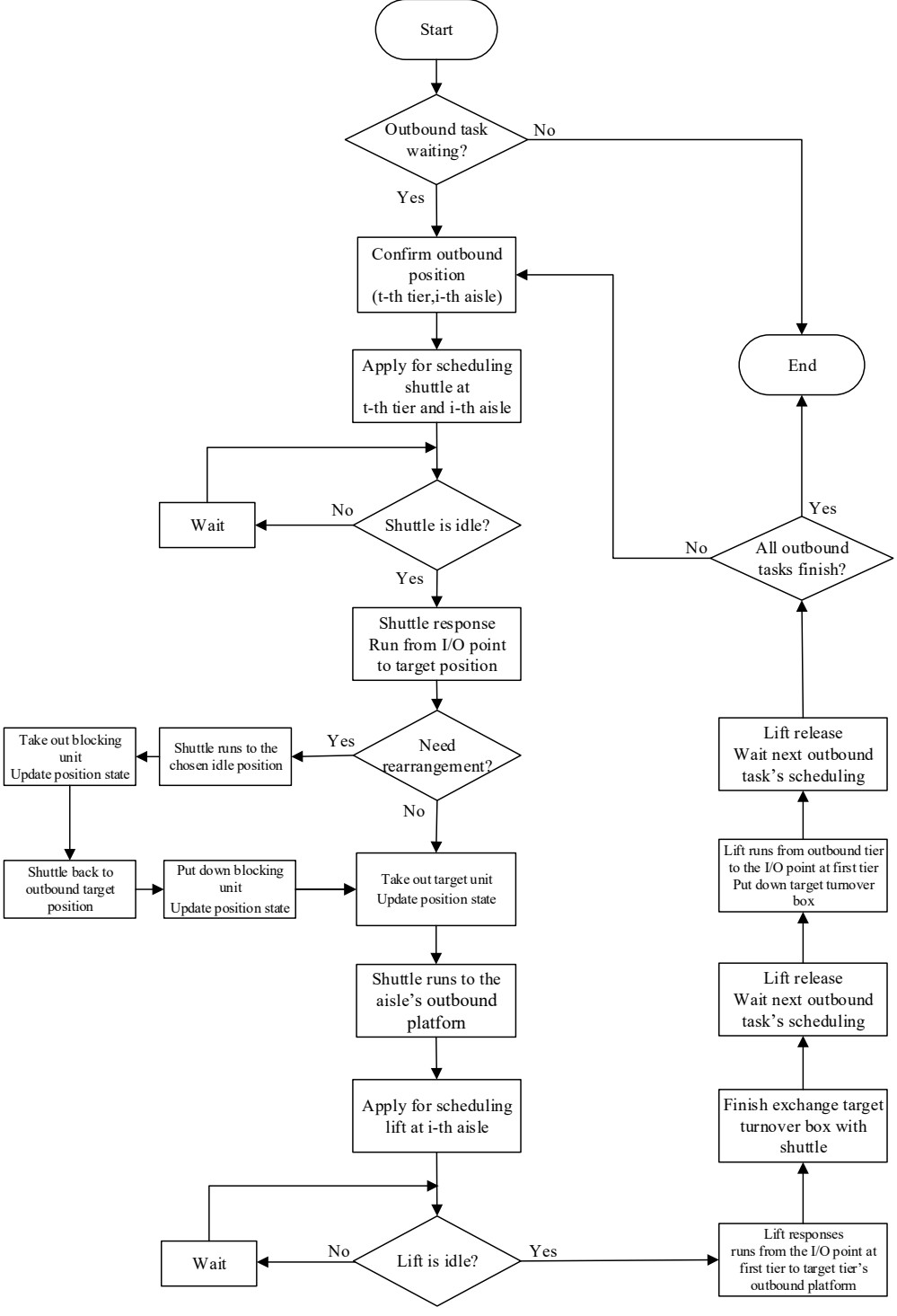

**Figure 3.** System operation process.

## 2. Background

At present, most of studies focus on the single-deep multi-tier shuttle system, and there is little research on the double-deep multi-tier shuttle system. Lerher [2] presented analytical travel time model for the computation of cycle times for double-deep shuttle-based storage and retrieval systems (SBS/RS), introduced the concept of cargo occupancy rate in the double-deep SBS/RS. According to Lerher [2], the double-deep SBS/RS has the advantages of fewer aisles, which results in a more efficient use of floor space. Since totes are stored in the first and in the second lane of the double-deep SBS/RS, chosen optimization is needed. Guo [3] puts forward two forms of chosen optimization problem of SBS/RS, but the parameter selection of genetic algorithm needs to be further studied. Yang [4] establishes the outbound time model of double-deep multi-tier shuttle system based on the occupancy rate of SKU position, but the shuttle outbound operation and inbound operation in SBS/RS are studied independently, and there is no overall study on the total time of the joint outbound operation of the two equipment. Mou [5] establishes the multi-objectives optimization model for operation performance based on parallel retrieval of shuttles and progressive transfer of the lift, but open queuing network still cannot fully conform to the reality. Xu et al. [1] takes the double-deep and double-load SBS/RS as the research objective, establishes the model of the system's outbound process, and compares it with the traditional single-deep and single-load multi-tier shuttle system. The results show that the efficiency of this system is significantly improved than single-deep and single-load multi-tier shuttle system. Although the single-deep multi-tier shuttle system does not have rearrangement process, the operation process of its shuttle and lift is similar to the double-deep multi-tier shuttle system, and its method of establishing time model can be used for reference.

Malmborg [6,7] studied autonomous vehicle storage and retrieval systems composed of shuttle and lift. He designed a conceptual method to analyze the shelf parameters and storage strategy's impact on system efficiency, but did not study the effect of the parameters on shuttle and lift. Based on stochastic storage and POSC (Point-of-Service-Completion) policy, Kuo [8] used an embedded queuing network model to analyze the average operation time of the shuttle and lift operating system, describing the shuttle's average waiting time and lift's service rate by iteration analysis, but how to analyze the system performance by mathematical model still needs further study. Zhang [9] proposed an open queuing network model in which the tasks were regarded as customers, shuttles and lifts were regarded as a group of parallel servers. However, it never involves the relationship between equipment dynamics and system performance. Gino [10] used the analysis model to evaluate the waiting time between the shuttle and the lift, and verifies it by simulation. But the model was established on the premise of fixed storage structure and equipment parameters. Compared with the open queuing network proposed by predecessors, both Cai [11,12] and Ekren [13] proposed to establish a semi-open queuing network to analyze the shuttle and lift operation system, but this system is only suitable for the serial operation system composed of shuttles and lifts.

Zou [14] established a Fork-join queuing network model for the access system composed of shuttle and lift, and used decomposition method to evaluate the system performance, but the double-deep SBS/RS is different from the single-deep SBS/RS, it needs to be rearrangement. Wang [15] proposed a time sequence mathematical model of task operation based on the movement characteristics of shuttles and lifts. Although the time sequence mathematical model has reference value, it also cannot adapt to the problem of rearrangement. Elena [16] developed queueing network model to estimate the performance of both single-tier and multi-tier shuttle-based compact systems, studied the best depth/width ratio of the system. In this paper, each tier and vertical model are modeled respectively, but in practice, the shuttle and lift run in parallel. Matej [17] established an independent optimization objective model of shuttle-based storage system of system operation time, cost and energy consumption. However, considering the amount of energy consumption, the problem of working time is ignored. Xu [18] based on the study of the dwell point policy of the 3D compact AS/RS, established the travel-time model of dual-command cycle.

In the above-mentioned literatures, there are few works on the task scheduling of double-deep multi-tier shuttle system, and the traditional model cannot adapt to double-deep multi-tier shuttle system. In order to improve the efficiency of the whole warehouse system, how to arrange different equipment and make it work together effectively is very important.

Recent literatures on optimization model of AS/RS have provided useful references for the optimization of double-deep multi-tier shuttle system, for example, Yang [4], Mou [5], and Liang [19]. Based on considering the kinematic characteristics of the double-deep multi-tier shuttle system, this paper firstly studies the cooperation between shuttles and the lifts. Starting from the outbound picking operation sequence, the outbound picking operation time model is established, so as to ultimately improve the outbound picking efficiency of the system. Secondly, the task scheduling model of outbound picking is established. Finally, the changing trend of outbound picking time with the columns' number of partitions is obtained by simulation, and the model is verified.

## 3. Methodology

### 3.1. Motion Characteristics of Equipment

Both shuttles and lifts in the system have acceleration $a$ and maximum velocity $v_{max}$. $D_c$ represents the critical distance when the equipment reaches maximum velocity $v_{max}$, $D$ represents the operating distance of the device and $v_0$ ($v_0 \leq v_{max}$) represents velocity of the device. When the distance $D > D_c$, the equipment can reach maximum velocity $v_{max}$ and the critical distance $D_c = \frac{v_{max}^2}{2a}$ can be obtained. In $D > D_c$, the operating process of the equipment (see Figure 4a) mainly includes three steps: (i) equipment accelerating from $0$ to $v_{max}$; (ii) equipment reaching a maximum velocity of $v_{max}$ and runs at a uniform velocity of $v_{max}$; (iii) equipment decelerating from $v_{max}$ until full stop. When the distance $D \leq D_c$, the equipment cannot reach the maximum velocity $v_{max}$. In $D \leq D_c$, the operating process of the equipment (see Figure 4b) mainly includes two steps: (i) equipment accelerating from $0$ to $v_0$; and (ii) equipment decelerating from $v_0$ until full stop.

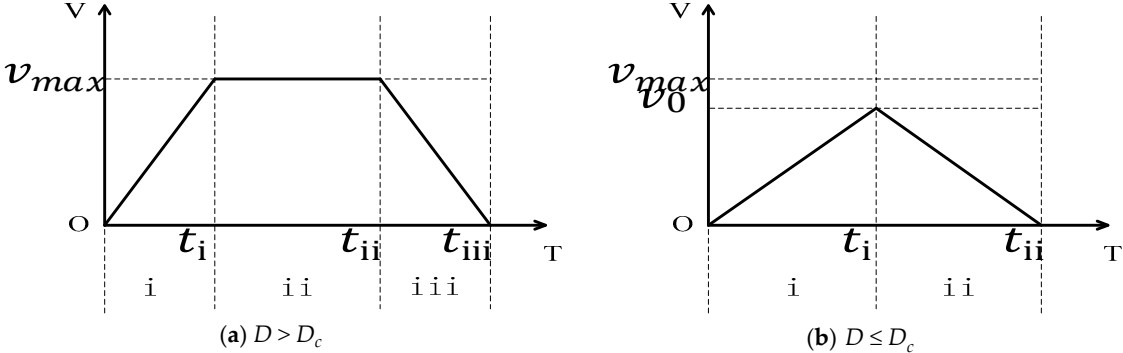

(**a**) $D > D_c$    (**b**) $D \leq D_c$

**Figure 4.** Operating process of the equipment.

According to the motion properties of equipment, when the equipment runs at the distance $D$, the running time $T_D$ can be calculated as Equations (1) and (2), respectively.

- When the distance $D > D_c$,

$$T_D = \frac{2v_{max}}{a} + \frac{D - 2D_c}{v_{max}} \tag{1}$$

- When the distance $D \leq D_c$,

$$T_D = \sqrt{\frac{D}{a}} \tag{2}$$

### 3.2. The Outbound Time Model

The operation mode of a double-deep multi-tier shuttle system is divided into two types: single operation and compound operation. Single operation mode refers to that the system only carries out outbound or inbound tasks in the single time window. Compound operation refers to the system's outbound operations and inbound operations proceeded simultaneously in the same time window. The double-deep multi-tier shuttle system gives priority to picking process and takes storage tasks as the auxiliary part. Therefore, outbound picking is its most important function. In the actual operation of the system, the priority of outbound operation is higher than inbound operation. Therefore, the research is based on the single operation mode.

Assuming that there are $M$ retrieval tasks in the system and these tasks are distributed in different aisles and different tiers. The number of aisles is $A$ (the number of lifts is also $A$), and the number of retrieval tasks in each aisle is $N_j$, where $N = \sum_{j=1}^{A} N_j$, $j \in \{1, 2, \ldots, A\}$. The batch of outbound tasks are all dealt in the same time window. The arrival time of the 1st retrieval task is 0. $N$ retrieval tasks' total time $T_N^{out}$ can be expressed as the maximum time to complete $N$ retrieval tasks, as shown in Equation (3).

$$T_N^{out} = max\left(time_i^{task}\right), \ i \in \{1, 2, \ldots, N\} \tag{3}$$

In Equation (3), $time_i^{task}$ represents the completion time of the *i-th* retrieval task in $N$ retrieval tasks.

In addition to Equation (3), $T_N^{out}$ can also be expressed by the maximum completion time of the $A$ lifts finish all retrieval tasks of each aisle, shown as Equation (4).

$$T_N^{out} = max\left(time_j^{lift}\right), \ j \in \{1, 2, \ldots, A\} \tag{4}$$

In Equation (4), $time_j^{lift}$ represents the completion time when the *j-th* lift completes all the retrieval tasks in its aisle. $time_j^{lift}$ can also express the total outbound time $T_{N_j}^{out}$ for the *j-th* lift to complete $N_j$ retrieval tasks of its aisle, and the total outbound time $T_N^{out}$ for the *n* tasks can be expressed as the Equation (5):

$$T_N^{out} = max\left(T_{N_j}^{out}\right), \ j \in \{1, 2, \ldots, A\} \tag{5}$$

$N$ retrieval tasks are distributed in different aisles and different tiers, which are finished by shuttles and lifts in serial or parallel operation. But for the $N_j$ retrieval tasks in *j-th* aisle, they are finished with shuttles' parallel retrieval operation and lift's serial operation; that is, the $N_j$ outbound tasks of the *j-th* aisle are completed by the lift serially. Therefore, the total outbound time $T_{N_j}^{out}$ of the *j-th* aisle can be obtained from the lift's serial operation, which is shown as Equation (6).

$$T_{N_j}^{out} = \sum_{k=1}^{N_j} T_k^{lift}, \ k \in \left\{1, 2, \ldots, N_j\right\} \tag{6}$$

In Equation (6), $T_k^{lift}$ represents the outbound operation time of the *k-th* retrieval task in *j-th* aisle.

From the lift's point of view, the time on the lift of the *k-th* outbound task can be divided into the waiting time $T_k^{l\_w}$ before the lift responds to the shuttle's dispatching, and the working time $T_k^{l\_r}$ after the lift responds to the shuttle's dispatching, that is Equation (7):

$$T_k^{lift} = T_k^{l\_w} + T_k^{l\_r} \tag{7}$$

Compositing Equations (6) and (7), $T_{N_j}^{out}$ can be expressed as Equation (8):

$$T_{N_j}^{out} = \sum_{k=1}^{N_j} \left(T_k^{l\_w} + T_k^{l\_r}\right), \ k \in \left\{1, 2, \ldots, N_j\right\} \tag{8}$$

For the *j-th* aisle, the serial operation time sequence of lift is shown in Figure 5.

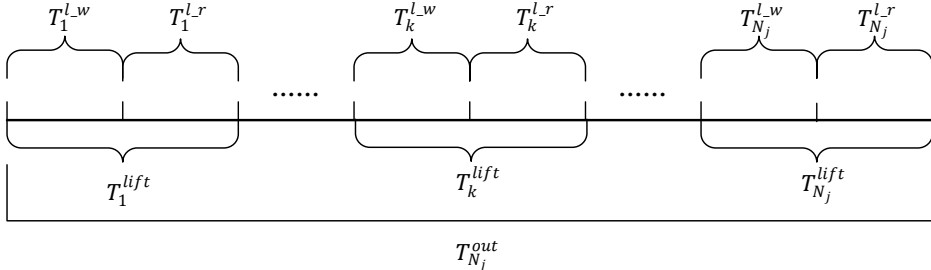

**Figure 5.** The serial operation time sequence of lift.

The following part analyzes detail of the lift's waiting time $T_k^{l\_w}$ and lift's operation time $T_k^{l\_r}$.

1. Lift's waiting time $T_k^{l\_w}$.

When the shuttle is applied for dispatching lift, lift may be in idle or busy state. When the lift is idle, the lift can immediately respond to the shuttle's dispatching request. So, the lift's waiting time is calculated from the completion of the last outbound task to the response of the shuttle's dispatching application. When the lift is busy, after completing the last outbound task, the lift immediately responds to the shuttle's dispatching and executes the outbound task. Thus, the lift is always in a busy state and there is no lift's waiting time. For more detailed analysis of lift waiting time $T_k^{l\_w}$, according to the outbound operation flow described above, we will analyze and defined time nodes of the *k-th* outbound task in the *j-th* aisle. The timing of the outbound task are shown in Table 1.

**Table 1.** Time nodes of the outbound task.

| Time | Corresponding Outbound Task Time Node |
|:---:|:---:|
| $t_k^0$ | Outbound task applies for dispatching shuttle |
| $t_k^1$ | Shuttle responds to outbound task scheduling application |
| $t_k^2$ | Shuttle runs to the outbound platform and applies for dispatching the lift |
| $t_k^3$ | Lift responses to shuttle scheduling application |
| $t_k^4$ | Lift runs to the corresponding outbound platform |
| $t_k^5$ | Lift and shuttle complete exchange turnover box (shuttle's release time) |
| $t_k^{task}(t_k^6)$ | Lift runs to the first tier's I/O point and puts down the turnover box (lift's release time). |

Combining with the analysis of the lift's waiting time and the time nodes of the outbound tasks described in the Table 1, it can be concluded that the lift's waiting time is divided into the following two cases:

(1)  when $t_k^2 > t_{k-1}^{task}$, the lift is idle, and there is a waiting time $T_k^{l\_w}$.

(2)  when $t_k^2 \leq t_{k-1}^{task}$, the lift is busy and the waiting time is $T_k^{l\_w} = 0$.

Set decision variable $L_k$ ($k \in \{1, 2, \cdots, N_j\}$) denotes whether the lift needs to wait at the time of the *k-th* outbound task of the *j-th* aisle in the system needing retrieval, such as shown in Equation (9).

$$L_k = \begin{cases} 1, & t_k^2 > t_{k-1}^{task} \\ 0, & t_k^2 \leq t_{k-1}^{task} \end{cases} \tag{9}$$

The time that lift waits for the shuttle, $T_k^{l\_w}$, can be expressed as Equation (10):

$$T_k^{l\_w} = L_k\left(t_k^2 > t_{k-1}^{task}\right), \ k \in \left\{1, 2, \ldots, N_j\right\} \tag{10}$$

The $t_{k-1}^{task}$ in Equation (10) can be expressed by the total time of the *1st k−1* outbound tasks in the aisle, namely Equation (11):

$$t_{k-1}^{task} = \sum_{m}^{k-1} T_m^{lift} \tag{11}$$

The $t_k^2$ in Equation (10) can be expressed as Equation (12) according to the time nodes table of outbound task.

$$t_k^2 = t_k^1 + T_k^{s\_r} \tag{12}$$

$T_k^{s\_r}$ in Equation (12) indicates the operation time after the shuttle responds to dispatching.

The shuttle's operation after responding to dispatching can be divided into two following situations. When the outbound task needs rearrangement, $T_k^{s\_r}$ includes two parts, i.e., taking the turnover box from the target position and rearranging the blocked turnover box. When the outbound task does not require rearrangement, the $T_k^{s\_r}$ only includes the operation time of fetching the turnover box from the target position without rearrangement time. Therefore, a decision variable $F_k$ is set to indicate whether the *k-th* task needs rearrangement or not, such as Equation (13). When the outbound task one and the task two are in the same batch of outbound tasks, the turnover box locations required for tasks one and two are determined, and the locations are located at different depths of the same container. The depth of the target turnover box of the task one is 1, and the depth of the target turnover box of the task two is 2. When the sequence of outgoing tasks is task one to task two, the rearrangement operation is need; when the sequence of outgoing tasks is task two to task one, there is no need to rearrangement.

$$F_k = \begin{cases} 1 & \text{Need rearrangement,} \\ 0 & \text{Not need rearrangement,} \end{cases} k \in \left\{ 1,\ 2,\ \ldots,\ N_j \right\} \tag{13}$$

Based on the above analysis, the operating time of shuttle $T_k^{s\_r}$ can be expressed as Equation (14):

$$T_k^{s\_r} = 2T_k^s + T^s + F_k T_k^{re'} \tag{14}$$

In Equation (14), $T_k^s$ denotes when the *k-th* outbound task is operating, the time the shuttle runs from the dwell point (i.e., the I/O station) to the target unit; $T_k^{re'}$ denotes the *k-th* outbound task's rearrangement time; and $T^s$ denotes the time the shuttle takes (puts) the turnover box, which is a constant.

According to Equations (10)–(12) and (14), we can get the waiting time of the lift $T_k^{l\_w}$ as Equation (15):

$$T_k^{l\_w} = L_k \left( t_k^1 + 2T_k^s + T^s + F_k T_k^{re'} - \sum_{m}^{k-1} T_m^{lift} \right), k \in \left\{ 1,\ 2,\ \ldots,\ N_j \right\} \tag{15}$$

2. Lift's operation time $T_k^{l\_r}$.

According to the outbound operation process, the lift's operation time $T_k^{l\_r}$ is mainly composed of four parts: (1) $T_k^{l1}$: The time lift runs from the dwell point (I/O point in the first tier) to the corresponding tier's outbound platform. (2) $T_k^{s\_l}$: The time to complete the transfer of the turnover box between the lift and the shuttle, which is a constant. (3) $T_k^{l1}$: The time lift delivers the target turnover box to the I/O point at first tier, which is the same as the value of $T_k^{l1}$. (4) $T^1$: The time it takes for the lift to put down the turnover box is same as the time $T^s$ for the shuttle to take or put a SKU and the time $T_k^{s\_l}$ for the lift and shuttle exchange the turnover box. It is only related to the hardware characteristics of the equipment and is constant. Therefore, lift's operation time $T_k^{l\_r}$ can be expressed as Equation (16).

$$T_k^{l\_r} = 2T_k^{l1} + T_k^{s\_l} + T^l, \; k \in \left\{1, \, 2, \, \ldots, \, N_j\right\} \tag{16}$$

Through the analysis of lift's waiting time and lift's operation time, the *k-th* outbound task's time $T_k^{lift}$ in the *j-th* aisle, as shown in Equation (17), can be obtained via Equations (7), (15) and (16).

$$T_k^{lift} = L_k\left(t_k^1 + 2T_k^s + T^s + F_k T_k^{re'} - \sum_{m}^{k-1} T_m^{lift}\right) + 2T_k^{l1} + T_k^{s_l} + T^1, \; k \in \left\{1, \, 2, \, \ldots, \, N_j\right\} \tag{17}$$

By synthesizing Equations (16) and (17), the total outbound time $T_{N_j}^{out}$ or the $N_j$-*th* outbound tasks in the *j-th* aisle is obtained as Equation (18):

$$T_{N_j}^{out} = \sum_{k=1}^{N_j}\left[L_k\left(t_k^1 + 2T_k^s + T^s + F_k T_k^{re'} - \sum_{m}^{k-1} T_m^{lift}\right) + 2T_k^{l1} + T_k^{s\_l} + T^1\right], \; k \in \left\{1, \, 2, \, \ldots, \, N_j\right\} \tag{18}$$

Through Equations (4), (5), and (18), the total outbound time $T_N^{out}$ of *N* outbound tasks in the system is Equation (19):

$$
\begin{aligned}
T_N^{out} = \; &max\left\{\sum_{k=1}^{N_j}\left[L_k\left(t_k^1 + 2T_k^s + T^s + F_k T_k^{re'} - \sum_{m}^{k-1} T_m^{lift}\right) + 2T_k^{l1} + T_k^{s\_l} + T^l\right]\right\}, \\
&k \in \left\{1, \, 2, \, \ldots, \, N_j\right\}, \; j \in \{1, \, 2, \, \ldots, \, A\}, \; m \in \{1, \, 2, \, \ldots, \, k-1\}, \; L_k, \, F_k \in \{0, \, 1\}
\end{aligned}
\tag{19}
$$

It can be seen from Equation (19) that the system efficiency can be improved by reducing the rearrangement distance $T_k^{re}$ when the equipment operation characteristics, task scheduling sequence and warehouse layout are fixed. In other words, transporting the block turnover box to the nearest spare location can reduce the distance of rearrangement.

### 3.3. Task Scheduling Model

The focus of task scheduling research is to optimize the processing sequence of work pieces, which is to order the processing tasks when production supply is limited [19]. The task scheduling problem of double-deep multi-tier shuttle system is similar with production scheduling problems. The retrieval tasks can be regarded as the "workpiece", and shuttles and lifts are seen as production devices. Figure 6 represent the "production process" of double-deep multi-tier shuttle system.

1. There are r assembly lines of parallel operations in the system and there are s shuttles and 1 retrieval lift for each assembly line; the shuttle of each line and r lines can do the operation simultaneously.
2. The production of each "workpiece" requires two steps. Step I is operated by shuttles; Step II is responsible by retrieval lifts. Once the "workpiece" is completed, the following shuttles and lifts are confirmed. The step II of all "workpieces" on assembly lines are accomplished by corresponding retrieval lifts.

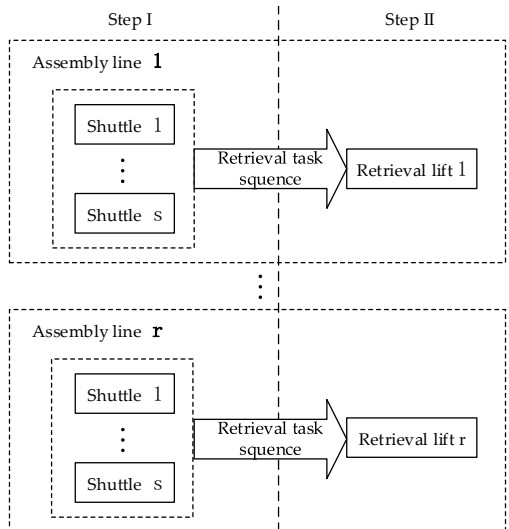

**Figure 6.** "Production process" of double-deep multi-tier shuttle system.

Generally speaking, the objective of task scheduling is to minimize the operation time for task scheduling optimization. Similarly, the objective for double-deep multi-tier shuttle system is to minimize the total time of retrieval tasks. From the Figure 6, since the operation of each assembly lines are simultaneous, the total production time is the maximum time when each assembly line finishes its total operation tasks. In order to minimize the total production time, it is essential to ensure the production time of each assembly line is the smallest. In double-deep multi-tier shuttle system, it is significant to guarantee the total retrieval time is minimum when each row finishes its all retrieval tasks. Assume that there are $M$ retrieval tasks in one row, and total retrieval time is $T_M^{out}$.

The objective, as shown in Equation (20), is to minimize the total time of accomplishing $M$ retrieval tasks in one row:

$$min T_M^{out} = \sum_{k=1}^{M} T_k^{lift} = \sum_{k=1}^{M} \left[ L_k \cdot \left( t_k^1 + 2T_k^s + T^s + F_k \cdot T_k^{re'} - \sum_{m=1}^{k-1} T_m^{lift} \right) + \left( 2T_k^{l1} + T^{s\_l} + T^l \right) \right] \qquad (20)$$

In this function, $T^s$, $T^{s\_l}$ and $T^l$ are all constants; $k \in \{1, 2, \ldots, M\}$; $m \in \{1, 2, \ldots, k-1\}$; $L_k \in \{0, 1\}$; $F_k \in \{0, 1\}$.

## 4. Model Solution Based on Modified Simulated Annealing Algorithms (SAA)

The task scheduling problem of double-deep multi-tier shuttle systems is similar to a vehicle routing problem with the goal of optimizing mission time subject to mission constraints on task precedence and agent capability [20]. This class of problems is NP-Completeness. While various heuristic algorithms search for the optimal solution of this kind of complex problem, they often cannot get the exact optimal solution, but they can get an approximate optimal solution more quickly. The solution time and accuracy of the approximate optimal solution are more consistent with the task scheduling in the actual situation [21]. At present, simulated annealing algorithm (SAA) and genetic algorithm (GA) are two widely used and effective algorithms in the field of task scheduling.

SAA is a general optimization algorithm and a global optimal algorithm. To some extent, SAA overcomes the local minimum defect and dependence on initial value in the optimization process of other algorithms, and has a strong global search performance [22]. At present, SAA has been widely used in many fields such as job shop scheduling, control science and so on. In this paper, the task scheduling optimization model in the double-deep multi-layer shuttle system minimizes exit time. Based on the above analysis, this paper uses SAA to solve the task scheduling model of the system.

*Application Design*

SAA is composed by solution domain, objective function, and initial solution. In what follows, we illustrate the solution domain and objective function respectively.

Solution domain: in double-deep multi-tier shuttle system, the scheduling combination sequence of all retrieval tasks compose a solution domain, which is represented as *Solution* = {$S_1$, $S_2$, ..., $S_U$}, and each $S_U$ consists of retrieval task sequence, *Task* = {$task_1$, $task_2$, ..., $task_M$}.

Objective function: As illustrated before, we set $T_M^{out}$ as objective function, the total time of accomplishing *M* retrieval tasks in one row.

To get better results, combined with specific task scheduling problems in double-deep multi-tier shuttle system, this paper utilized SAA and modified SAA with local search for optimization. The characteristics of simulated annealing algorithm require higher initial temperature and slower temperature drop rate, and if the parameters are improperly selected, it will fall into local optimization. Therefore, to avoid this situation, a local search method, the local search method of individual exchange are introduced into the algorithm. The specific process and relevant parameter settings are listed as follows:

(a) Initialization parameter control.

  i. Initial temperature, $T_0$: $T_0$ determines the annealing pace directly. If higher initial temperature is set, it is easier to get optimized result. However, when the initial temperature value is set too high, the iteration time of algorithms increases.

  ii. Final temperature, *Te*: *Te* is the terminal condition of the algorithm. When *T* = *Te*, the algorithm terminates.

  iii. Number of iterations for each *T*, *L*: *L* is the length of Markov Chain, which is the number of iteration under fixed temperature; when the temperature attenuation coefficient is determined in advance, setting the magnitude of *L* makes the temperature value approach to quasi equilibrium. Normally, *L* is less than 1000.

  iv. Temperature attenuation coefficient, $\alpha$: attenuation coefficient is 1 or slightly less than 1.

(b) Generate initial solution *X0*, choose a retrieval task sequence Su from solution domain Solution as the initial solution; calculate the objective function *E(X0)*, which is the total time of retrieval tasks under corresponding task sequence.

(c) For current number of iterations *k* = 1, 2, ..., *L*, operate step (d) to (g).

(d) Generate new solution *X'*; generate a new retrieval task sequence by certain method; and calculate the objective function *E(X')*, total time of retrieval tasks, under new sequence.

The new solution is to generate new retrieval tasks sequences by the local search method of individual exchange [23]. The steps are: (1): Randomly select two positions x and y in current sequence, exchange the retrieval tasks of x and y, then get a new retrieval task sequence. (2): Calculate the adaptive value based on new retrieval task sequence (adaptive value is the reciprocal of total time of retrieval tasks). If the adaptive value is larger than the value of old sequence, then replace it and operate. If not, then use the adaptive value of old sequence and operate. (3): Go back to step (1) until the algorithm termination. The condition to terminate algorithm is to set certain number of local searches.

(e) Calculate increment ∆*E*, time difference between retrieval tasks.

(f) Metropolis Guidelines. If ∆*E* < 0, then accept X' as the current retrieval task sequence; if ∆*E* > 0, then accept X' as the current retrieval task sequence with the probability exp (−∆*E*/*T*).

(g) If *k* ≤ *L*, go to step (d); if not, go to step (h).

(h) If current temperature *Tc* ≤ *Te*, export the current retrieval task sequence as the best operation sequence, and terminate algorithm; if not, decrease the temperature value *Tc*, *Tc* = $\alpha$ *Tc* and go to step (d).

Figure 7 represents the flow chart of the algorithm.

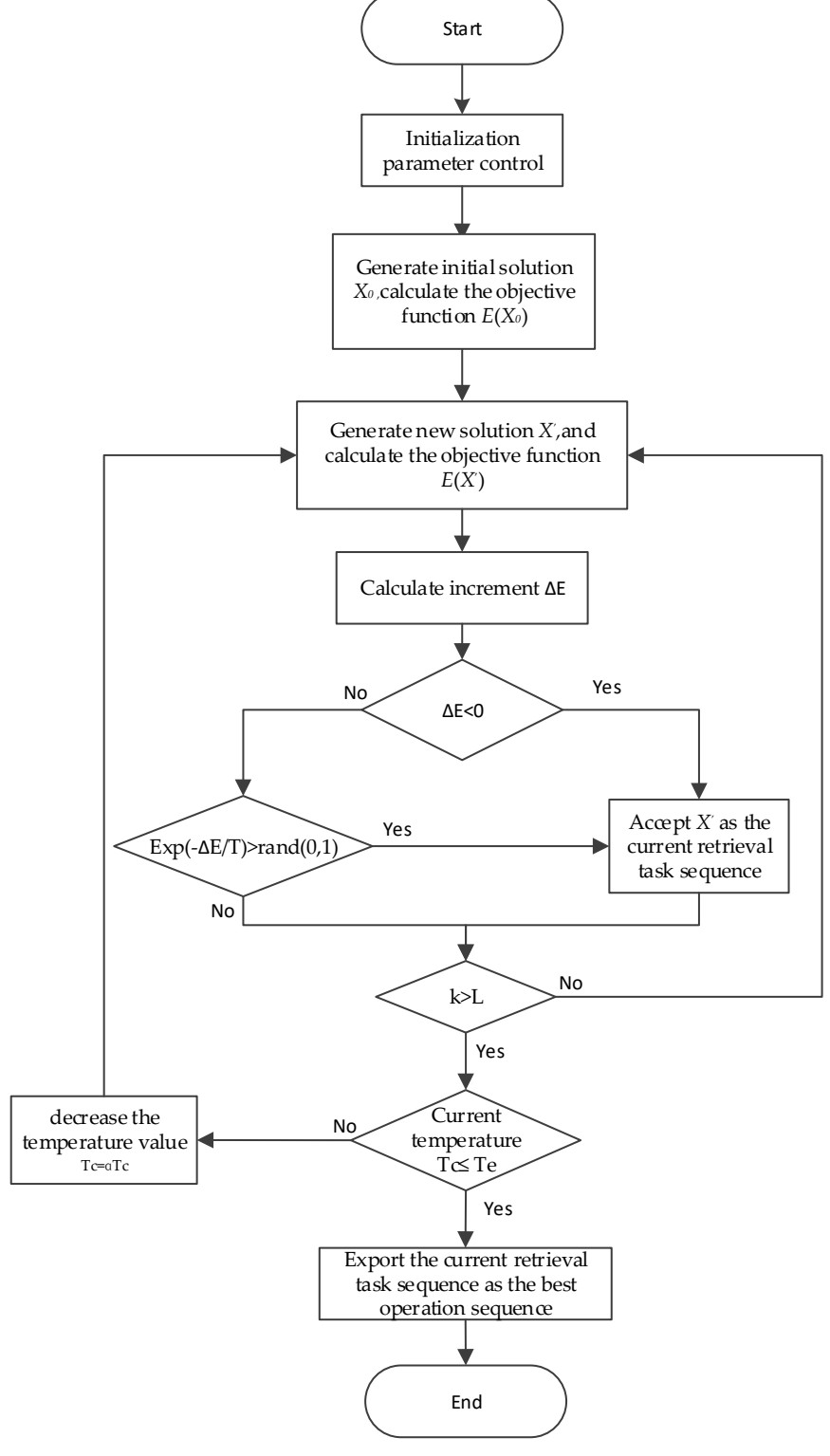

**Figure 7.** Simulated annealing algorithm flow chart.

## 5. Case Study

This simulation example data is from a clothing brand manufacturer's finished product warehouse. Its system layout and device parameters are shown in Table 2.

**Table 2.** System layout and device parameter.

| Hardware | Parameter | Parameter Values |
|---|---|---|
| Goods shelves | Number of tiers | 5 tiers |
| | Number of rows | 30 rows |
| | Number of aisles | 8 |
| | Tier height | 0.8 m |
| | Column width | 0.5 m |
| Shuttle | Maximum speed | 2 m/s |
| | Acceleration | 1 m/s$^2$ |
| | Take (put) time | 1.5 s |
| Lift | Maximum speed | 1 m/s |
| | Acceleration | 0.5 m/s$^2$ |
| | Take (put) time | 1.5 s |
| Other | Turnover box transfer time | 3 s |

The simulation experiments are illustrated as follows:

1. The aisles' outbound tasks work in parallel and are independent from each other, thus we select one aisle for simulation analysis.
2. SKU positions are random distributed, and position's occupancy rate is 0.6, initialization data are stored in Excel.
3. The number of partition columns x is set to be 8, and get the outbound time of different partition sizes under the same task sequence.

The aim of the case study is to optimize the retrieval task scheduling sequence and confirm the best optimization result. The retrieval tasks on each aisle are independent. The case study chose an aisle with 50 and 100 retrieval tasks, respectively. Table A1 lists the information of partial retrieval tasks, which includes SKU number and location coordinates. The coordinate is represented by (tier, row, depth). The range of tier is 1 to 5; the range of row is 1 to 30; and the range of depth is 1 to 4. Specifically, value of 1 or 4 represents the depth is 2, and value of 2 or 3 means the depth is 1.

The authors use MATLAB 2013A for simulation experiments on Intel(R) Core$^{TM}$ i5-2430M CPU @ 2.50GHz 2.40GHz dual-core processor, 4.00 GB PC. According to Chen and Shahandashti [24], we set the initial temperature $T_0$ = 2000 and number of iterations $L$ = 50. Combined with explained parameters before, the initial parameter setting of SAA for the case is listed in Table 3.

**Table 3.** Initial parameters setting of Simulated Annealing Algorithm (SAA).

| Parameter Setting | Initial Temperature $T_0$ | Final Temperature $T_e$ | No. of Iteration $L$ | Temperature Attenuation Coefficient, $\alpha$ |
|---|---|---|---|---|
| Initial Values | 2000 | 0.001 | 50 | 0.98 |

Since the SKU units are different in one batch of tasks, we numbered to represent each retrieval tasks. Then, we randomly selected two sets of data with different number of tasks to optimize. The results are shown in Tables A2 and A3. When the number of tasks is 50, the retrieval task sequences before and after optimization by modified SAA is listed in Table A2. Before optimization, the retrieval

operation time is 368.35 s; after optimization, the retrieval operation time is 326.85 s. Comparison between retrieval task scheduling before and after optimization is shown in Figure 8. The retrieval operation efficiency improved 11.3%.

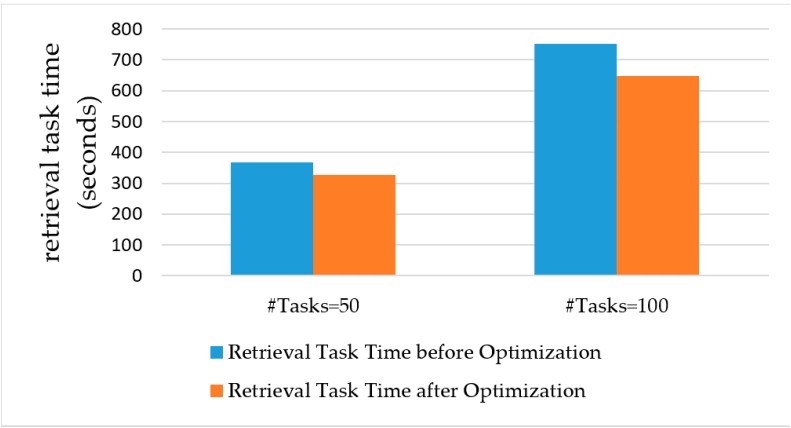

**Figure 8.** Comparison between retrieval task scheduling before and after optimization.

In the analysis of case study, to ensure the reliability of the data, we randomly selected multiple sets of data when task is 50 or 100, respectively. And the average of optimized retrieval efficiency improved 10.9% and 12.3%, respectively. The result verified that the modified SAA can solve the task scheduling problem for double-deep multi-tier shuttle system efficiently and get the optimized sequence.

## 6. Conclusions

The paper focuses on the optimization of retrieval task scheduling problem of double-deep multi-tier shuttle system. The rational retrieval task sequence can decrease the number of rearrangement of goods and shorten waiting time of lifts. Thus, the overall retrieval operation efficiency will be improved. The task scheduling of a warehouse system can be identified as all outbound tasks within a time window size. The paper firstly introduced the task scheduling problem for the multi-tier shuttle system. In addition, inspired from the production scheduling problems, we set to minimize the total time of accomplishing a batch of retrieval tasks in one row as objective function, used the modified SAA, and solved the task scheduling problem. In conclusion, we verified the efficiency of proposed model and algorithms by case studies.

## 7. Limitations

In this study, we make several assumptions that may be investigated in future research. For example, the outbound operation time model in this paper is established in the case of only considering the outbound task. In other word, the system only considers the outbound task and not the inbound task in a time window. In the actual operation process, the system carries out both outbound and inbound operation in a time window, which increases the complexity of system task scheduling. However, it has more practical significance for the research of task scheduling in compound job mode.

There have been many theoretical studies on cargo space allocation in AS/RS and other systems, but there are great differences between the double-deep multi-shuttle system and the above-mentioned systems. Therefore, the theoretical research of cargo space allocation in other systems cannot be fully applied to the double-deep multi-shuttle system. In particular, how to allocate the same shelf depth and different cargo space, in order to improve the efficiency of the system, is worthy of further study.

**Author Contributions:** Conceptualization, Y.W. and R.Z.; Data curation, X.Z.; Formal analysis, Y.W.; Investigation, R.Z., H.L. and Z.L.; Methodology, Y.W.; Resources, Y.W. and Z.L.; Supervision, Y.W.; Validation, Y.W., H.L. and X.Z.; Writing—original draft, R.Z. and X.Z.; Writing—review and editing, H.L.

**Funding:** This research was funded by the Shandong Province Key Research Funds, grant number 2017GGX60103, the Fundamental Research Funds for Shandong University, grant number 2018JC035, and the Shandong Provincial Natural Science Foundation, grant number ZR2017MF014.

**Conflicts of Interest:** The authors declare no conflict of interest.

## Appendix A

**Table A1.** List of partial retrieval tasks.

| SKU No. | Coordinates | SKU No. | Coordinates |
|---|---|---|---|
| 114 | (1, 14, 1) | 360 | (3, 19, 3) |
| 131 | (1, 1, 4) | 364 | (3, 19, 3) |
| 142 | (1, 12, 4) | 366 | (3, 23, 3) |
| 148 | (1, 18, 4) | 367 | (3, 27, 2) |
| 150 | (1, 20, 4) | 369 | (3, 29, 2) |
| 158 | (1, 28, 4) | 378 | (4, 8, 1) |
| 159 | (1, 29, 4) | 389 | (4, 19, 1) |
| 171 | (1, 11, 2) | 390 | (4, 20, 1) |
| 179 | (1, 19, 2) | 392 | (4, 22, 1) |
| 185 | (1, 25, 2) | 394 | (4, 24, 1) |
| 197 | (2, 7, 1) | 409 | (4, 9, 4) |
| 224 | (2, 4, 4) | 419 | (4, 19, 4) |
| 227 | (2, 7, 4) | 430 | (4, 30, 4) |
| 234 | (2, 14, 4) | 434 | (4, 3, 3) |
| 239 | (2, 19, 4) | 446 | (4, 15, 3) |
| 243 | (2, 23, 4) | 454 | (4, 23, 3) |
| 262 | (2, 11, 3) | 465 | (5, 5, 1) |
| 266 | (2, 15, 3) | 482 | (5, 22, 1) |
| 292 | (2, 7, 1) | 493 | (5, 3, 4) |
| 299 | (3, 19, 1) | 518 | (5, 28, 4) |
| 325 | (3, 15, 4) | 523 | (5, 3, 2) |
| 326 | (3, 16, 4) | 527 | (5, 7, 2) |
| 335 | (3, 25, 4) | 530 | (5, 9, 3) |
| 349 | (3, 9, 2) | 538 | (5, 17, 3) |
| 353 | (3, 13, 2) | 543 | (5, 23, 2) |

**Table A2.** The list of retrieval tasks before and after optimization when Task = 50.

| | |
|---|---|
| Retrieval Task Sequence before Optimization | 527, 227, 392, 224, 325, 434, 493, 299, 262, 114, 446, 171, 543, 197, 239, 131, 148, 518, 266, 454, 465, 409, 142, 538, 292, 150, 179, 349, 366, 353, 243, 394, 523, 360, 158, 430, 389, 530, 378, 419, 185, 159, 369, 364, 482, 234, 335, 367, 326, 390 |
| Retrieval Task Sequence after Optimization | 523, 360, 378, 419, 366, 434, 430, 114, 530, 349, 171, 446, 142, 326, 299, 158, 150, 224, 266, 148, 353, 367, 538, 185, 262, 454, 392, 518, 543, 131, 325, 394, 227, 292, 527, 409, 389, 493, 243, 335, 239, 159, 369, 234, 482, 364, 179, 465, 197, 390 |

Table A3. shows the list of retrieval tasks before and after optimization when Task = 100. Before optimization, the retrieval operation time is 752.1 s; after optimization, the retrieval operation time is 647.6 s. The retrieval operation efficiency improved 13.9%.

**Table A3.** The list of retrieval tasks before and after optimization when Task = 100.

| | |
|---|---|
| Retrieval Task Sequence before Optimization | 269, 549, 200, 531, 463, 427, 160, 335, 297, 336, 318, 208, 342, 141, 166, 246, 542, 152, 450, 101, 477, 279, 364, 504, 529, 164, 354, 276, 483, 116, 221, 151, 255, 333, 111, 174, 480, 454, 192, 330, 471, 370, 133, 134, 547, 366, 110, 433, 247, 239, 441, 114, 113, 278, 115, 193, 327, 513, 190, 422, 313, 534, 209, 406, 348, 444, 385, 413, 419, 407, 533, 535, 462, 155, 369, 226, 211, 425, 410, 219, 288, 323, 106, 307, 503, 537, 171, 379, 484, 223, 181, 527, 458, 150, 146, 448, 351, 486, 207, 319 |
| Retrieval Task Sequence after Optimization | 336, 164, 433, 549, 133, 116, 531, 297, 110, 342, 101, 221, 192, 208, 454, 247, 318, 239, 354, 279, 483, 529, 364, 547, 463, 480, 160, 152, 477, 174, 542, 141, 366, 151, 111, 370, 269, 276, 335, 166, 471, 134, 450, 246, 255, 504, 333, 200, 330, 427, 441, 288, 114, 113, 278, 115, 193, 219, 207, 327, 513, 190, 351, 448, 223, 422, 155, 313, 534, 209, 406, 533, 348, 458, 444, 385, 413, 419, 407, 535, 462, 369, 226, 211, 425, 410, 323, 106, 307, 503, 537, 171, 379, 484, 181, 527, 150, 146, 486, 319 |

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
