# Peer review of "Task Scheduling Model of Double-Deep Multi-Tier Shuttle System"

_processes, doi:10.3390/pr7090604_

Round 1
Reviewer 1 Report
This paper proposes a new type of part-to-picker storage system which is the double-deep multi-tier shuttle system. The proposed system is described in details and hence easy to be understood and appreciated. Then, the proposed system is optimised used SA and the optimisation is evaluated using a scenario related to a clothing warehouse.
The paper is relatively well written and easy to follow, despite the rather large number of formulas in section 2. This is thanks to the helpful illustrations and good description. This part requires a rather low number of improvements. Some of them are highlighted in the attached pdf file.
In the paper, there is no separate 'Related work' section and the prior works have been described in the introduction (section 1). In my option, this survey is rather superficial; the highlights of all the referred works are summarised in one sentence, but no discussion of this research is provided. Hence, the Author's conclusion that 'the theoretical research is very scarce' in line 98 is not obvious. Please, comment on the related works a bit more than the motivation for your research is more clear.
As written above, the description of the system model is quite well done. Some minor flaws are highlighted in the attached pdf. Similarly, the task scheduling model is described well enough. The main problem is, in my opinion, related to the optimisation part and the evaluation. The general introduction to SA is not convincing and a number of mistakes has been spotted (please, see again the attached pdf). Anyway, as the SA is a well-known concept, this section, after improvement, can be treated as sufficient as well. However, section 6 requires a significant improvement. Firstly, it evaluates only the SA-based optimisation. But the main contribution of the paper, namely, a new type of part-to-picker storage system, is not evaluated. So, the benefits of applying the double-deep multi-tier shuttle system is unclear. Please, provide a sufficient comparison with state-of-the-art part-to-picker storage systems, as otherwise the value of your proposal is impossible to be assessed. Also, SA-based optimisation, despite being a rather insignificant contribution (we all know that optimisation metaheuristics are likely to improve the schedule), is evaluated rather briefly. Only two scenarios have been analysed (50 and 100 tasks) and, if I've understood it correctly, the optimisation has been performed only once. As SA is not deterministic, the experiment needs to be performed a number of times to achieve the statistical significance.
Please, improve the paper presentation in terms of the language, punctuation and denotations. Some issues are highlighted in the pdf, but there are much more, especially in the second half of the paper.

Reviewer 2 Report
The paper presents the new approach to optimise the scheduling of double-deep multi-tier shuttle system using Simulated Annealing Algorithm (SAA). The research conducted identifies the new opportunities for improving the efficiency of warehousing and production line using new developments. The outcome of the study seems interesting. However, the paper requires significant improvements for being published.
Introduction
The paper started with the introduction, which looks more like a literature review than really introducing the topic or the idea of why this research is needed. Furthermore, the introduction has various flaws and requires major corrections with particular attention to the schematic flow of information and tone. Mistakes in terms of text also need corrections.
For example :
The term SKU is not introduced in the first place but used in the whole paper. line 42, the claims are not backed by references Line 46 the layout could be complemented with the image to make it easy to understand paragraph 3 could be divided into multiple parts; also, the reference format is not correct when referring to authors name at the starting of the sentences. Line 55 then and this? confusing line 58 again use of “then”, flow is not correct Line 62 the policies proposed by who? it is tough to follow the arguments made in the paper Line 64 spelling mistake “Shen” Line 71 contains the text which actually should be starting the introduction section to make the flow more easy to understand A few examples could back line 101, recent literature.The flow of text makes it unclear to why the research is conducted at the first place, what is the need, why is it essential to refer to some of the works presented, what gaps are remaining. The text is there, but restructuring is required to make these question more apparent to the readers.
I recommend authors to create separate sections: Introduction and Background (or Previous work)
System Description
This section mostly belongs to the part of introduction then being part after the introduction. I recommend the restructuring by moving this section before the literature review to make clear to the reader what is the system all about. Furthermore, the text needs more clarity. For example, corresponding I/O? How many I/O are there? Based on the image its one ?? need better wording or explanation.
The flow chart used is very useful.
System Travel Time Analysis
Authors should reconsider the structuring of the paper. May be creating a section “Methodology” can incorporate several sections in the current state. Attention needs to be given on the text also. For example, line 142 and 146; ”The operating process of the equipment” is repeated but referring to a different image, hence making the whole flow and understanding of text confusing. Equation 7 and 8 need to be revisited for any corrections.
I suggest the authors to consider the same for Section 4 and 5, maybe with restructuring.
Examples :
Acronym NP-Completeness is not explained before it was used. Please discuss in detail the difference between SAA and modified SAA? What paper present is SAA with modifications or its a different scope for SAA? Few variable names need formula considerations, some time they are in italics in the text some times not. Please synchronise the notions. What are step 4 and 8 are they referring to your roman numbers or something else? Please consider referring to what you actually present in the paper.Case study
I recommend authors to please proofread the language and the notions of the major point discussed in the paper with grammatical errors. For instance, authors say in the table “Number of columns ” and line 336 the range of rows, line 332 “cast”?
It would be intuitive if authors can provide the annealing energy transition plot too, along with the graphs already in the paper.
The authors have not discussed the limitations, outlook, discussion concerning other research work and concrete conclusion of the finding apart from case studies’ outcome of the present research work. I suggest authors to add separate sections: Limitations, Outlook, Discussion.
Overall attention is required, especially notions of research, grammar check and schematic flow of text need to be considered.
Round 2
Reviewer 1 Report
The paper has been improved since its previous version and, in its current version, I have no more negative remarks concerning it.
Reviewer 2 Report
The paper is well improved considering the reviews. However, a extensive proofread is still recommended.